# Observations of Ground-Level Ozone Concentration Gradients Perpendicular to the Lake Ontario Shoreline

Yao Yan Huang[1, 2], D. James Donaldson[1, 2]

[1]Department of Chemistry, University of Toronto, Toronto, M5S 3H6, Canada

[2]Department of Physical & Environmental Sciences, University of Toronto Scarborough, Toronto, M1C 1A4, Canada

*Correspondence to*: D. James Donaldson (james.donaldson@utoronto.ca)

**Abstract.** Ground-level ozone (O3) is a secondary air pollutant that has harmful effects on human and ecosystem health. Close to larger bodies of water, the well-known sea- (or lake)-breeze phenomenon plays a role in regulating ground level ozone levels. An observed lake-edge removal effect, where ozone concentration decreases within the first 500 m to 1 km

perpendicular to the lake, is thought to be related to the lake-breeze circulation as well as several dilution and removal pathways. A field campaign was conducted in summer 2022 and winter 2023 in two locations on the north shore of Lake Ontario: the urban centre of Toronto, and suburban Oshawa, some 50 km east, to assess how the local environment and season effects the lake-edge removal effect. Ozone, wind speed, and wind direction were measured on 6-7 different days in each season and city along transects perpendicular to Lake Ontario's shoreline. A consistent negative linear relationship between

ozone concentration and distance from shore over the first 500 m (i.e. a lake-edge removal effect) was observed in both cities and both seasons. The ozone gradient changed in Oshawa from -23.5 ± 8.5 (1 standard deviation) ppb/km in summer to -8.1 ± 5.1 ppb/km in winter. The slope remained consistent in Toronto at -15.4 ± 6.7 ppb/km in summer and -16.7 ± 7.3 ppb/km in winter. The year-round observation of an ozone gradient and lake-edge removal effect suggests that there is an inherent "baseline" ozone concentration gradient at the lake edge, caused by the dilution effect of the rapid increase of the boundary

layer there. This gradient is seen to be enhanced in the summer and dependent on local effects such as lake-breeze circulation and increased deposition to vegetation.

## 1 Introduction

Ground-level ozone ($O_3$) is a secondary air pollutant which, at higher concentrations, has harmful effects on human health (Nuvolone et al., 2018) and ecosystems (Grulke & Heath, 2020). This has motivated considerable efforts at understanding its

production, chemistry, and removal from the lower atmosphere. Sea- and lake-breeze circulations are a well-established meteorological phenomenon that plays a role in regulating ozone levels in nearshore environments. Sea- and lake breezes are driven by heat flux differences between land and water, due to the difference in their respective heat capacities. A pressure difference is generated between the warmer air over land and cooler air over water in the daytime, with airflow near the shore moving landward to replace the lofting, warmer air over land. The opposite effect occurs at nighttime, as the water holds heat

better after the sun is no longer directly heating the surface (Sills et al., 2011). An offshore "land-breeze" moves ozone precursors emitted during the night and early morning from the land to the lake where they are trapped in a shallow layer of cool and stable air over the water. The increase in fresh emissions and the lower deposition rates over water than land builds up the over-water ozone and ozone precursor concentration during the night and early morning (Levy et al., 2010). During the day, the wind direction switches, and an onshore "lake-breeze" transports the newly produced $O_3$ and $O_3$ precursors inland in

the afternoon and early evenings (Dye et al., 1995). This process is illustrated in Fig. 1. Clear calm skies are favourable for lake-breeze circulation formation and most develop in the summer from May to September in the northern hemisphere when the temperature differences are most pronounced (Wentworth et al., 2015).

Studies of the effect of sea-and lake-breeze circulation on ozone pollution have been conducted since the 1970s (Blumenthal et al., 1978; Chung, 1977; Doak et al., 2021; Dye et al., 1995; Lennartson & Schwartz, 2002; Lyons & Cole, 1976; Makar et

al., 2010; Stauffer & Thompson, 2015). Measurements have been made by ferry (Cleary et al., 2015), aircraft campaigns (Levy et al., 2010), and with ground data (Lyons & Cole, 1976) to model and understand the local scale flow pattern. Elevated levels of ozone pollution over water and coastal areas have been studied globally in areas such as Chesapeake Bay, United States (Stauffer & Thompson, 2015), Naples, Italy (Finardi et al., 2018), and Hanzhou, China (Han et al., 2023). The relationship between ozone pollution and lake-breeze circulation has been studied in the Norther American Great Lakes region such as

Lake Michigan (Cleary, De Boer, et al., 2022; Cleary, Dickens, et al., 2022; Cleary et al., 2015; Lennartson & Schwartz, 2002), Lake Erie (Levy et al., 2010), Lake Ontario (Wentworth et al., 2015), and globally, such as in Lake Taihu in China (L. Zhang et al., 2017). The concentration differences between lake and land are most pronounced below 200 m altitude (Levy et al., 2010; Tirado et al., 2023). In the afternoon, the highest ozone concentrations over land are seen at higher altitudes above the boundary layer (Tirado et al., 2023).

The advection of ozone-rich air inland by sea-and lake-breezes can lead to high $O_3$ at nearshore land regions. Lake-breezes have been reported to penetrate over 100 km inland in the Great Lakes region (Sills et al., 2011) with ozone enhancement effects observed typically up to 30-50 km inland (Lennartson & Schwartz, 2002; Lyons & Cole, 1976). Correlations with lake-breeze have been observed for daily $O_3$ maxima (Wentworth et al., 2015), the occurrence of secondary daily ozone peaks (Lyons & Cole, 1976), and ozone exceedances in nearshore sites in time and space following the movement of the lake breeze

front (Lennartson & Schwartz, 2002). Average daytime $O_3$ concentrations in the Greater Toronto Area were 42-49% higher when lake breezes were present, with mixing ratios at least 30 ppb higher in sites within the circulation than outside, despite similar meteorological conditions and regional synoptical regimes (Wentworth et al., 2015).

Surface gradients of ozone perpendicular to the shoreline over the adjacent land have also been correlated with lake breeze circulation. Ozone is typically highest near the water and decreases inland(Blanchard & Aherne, 2019; Cleary, Dickens, et al.,

2022; Lennartson & Schwartz, 2002). The Lake Michigan Ozone Study 2017 investigated coastal ozone gradients and measured ozone amounts of 81.4 ppb and 87.4 ppb at the beginning and end of the transect near the shore, and 57.4 ± 1.6 ppb

(error bars represent 1 standard deviation) at distances more than 4.1 km (Stanier et al., 2021). A recent study measured surface ozone gradients of 18 ppb km$^{-1}$ and 15 ppb km$^{-1}$ respectively on the south and north shores of Long Island, New York, United States in around 2 km distance (J. Zhang et al., 2020). The steepness of the ozone gradient is also correlated with the strength

of the lake-breeze, as nearshore lake breeze events are reported to give rise to a steeper ozone gradient than inland lake-breezes that penetrate further (Cleary, Dickens, et al., 2022).

A recent field campaign in Sandbanks Provincial Park, located on the north shore of Lake Ontario about 220 km east-northeast of Toronto, during summer 2018 found that surface ozone concentration decreases sharply within the first 500 m to 1 km perpendicular to the lake, in addition to a shallower gradient extending beyond 1 km from the lakeshore (Blanchard & Aherne,

2019). Sites < 500 m from shore had an ozone gradient of -37.6 ppb/km, ($R^2 = 0.72$) while sites > 500 m from shore had a gradient of -4 ppb/km with distance from shore with a weaker correlation ($R^2 = 0.17$). The authors called this steeper gradient a lake-edge removal effect and hypothesized that polluted air masses transported onshore experienced increased removal effects as they interacted with fragmented vegetation, similar to the forest-edge effect (Karlsson et al., 2006). In addition, they hypothesized that sand dunes running parallel to the coast may generate turbulent air flow, leading to additional mixing with

vegetation and subsequent removal (Blanchard & Aherne, 2019).

A recent study shows that even high-resolution ozone air quality models still show bias that can be attributed to transport and lake breeze errors (Abdi-Oskouei et al., 2020). A better understanding of this edge effect can improve models and better inform policy related to land use and human health. The present measurement report aims to provide a better understanding of this steep nearshore ozone gradient, that will be referred to in the following as a lake-edge removal effect. A field campaign was

conducted from June-August 2022 (summer) and December 2022-February 2023 (winter), with some follow up measurements in early summer 2023 to investigate the short-term reproducibility, seasonal dependence, and effect of land local forms on the lake-edge removal effect.

## 2 Method

### 2.1 Study Sites

The field campaign consisted of measurements from June-August 2022 (summer) and December 2022-February 2023 (winter) in Toronto, a built-up urban environment, and in Oshawa, a more suburban setting located about 50 km east of Toronto. Sampling sites within Toronto and Oshawa were chosen to evaluate the spatial variation of ground-level ozone along a linear transect perpendicular to the lake in urban and suburban areas respectively. Both cities have stations in Ontario's Ambient Air Monitoring Network that measure $O_3$, $NO_2$, $NO_x$, $SO_2$ and $PM_{2.5}$, providing a basis for comparison and calibration points.

These more urban locations were selected as a direct contrast to the previous study by Blanchard & Aherne (2019) that sampled in Sandbanks Provincial Park, a public beach and forested park outside Belleville Ontario. The cities are also close enough

that differences in regional background ozone levels should not interfere with nearshore measurements. Both cities were sampled again twice in June-July 2023 (summer) to confirm observations were a persistent yearly phenomenon. Six sampling sites lying between the two cities were also selected to evaluate ozone levels parallel to the lake and sub-regional ozone differences.

## 2.2 Toronto Sites

The Toronto lakeshore site lies in the harbourfront area of downtown Toronto. Figure 2 displays the sampling locations chosen. The major features near the shore consist of a concrete and wood boardwalk, docks, sidewalks, roads, and mid to high-rise buildings. High vehicular and foot traffic occur along major roads. The Gardiner Expressway runs parallel to the shoreline within the region. Some boat and ferry traffic operates from spring-fall.

Thirteen locations ranging from 3580 m to 15 m from shore were sampled during 6 days in the summer and 7 days during the winter measurement period (see Table S1). The weather in summer was mostly consistent, ranging from sunny to cloudy with mean daily temperatures from 19.0°C to 26.3°C. In winter, only locations within 1 km of the shore were sampled, location IDs 29 to 37 (see Fig. 2), spanning a distance of 820 m to 15 m from shore. One additional location (ID 34) was added during the winter period that were not sampled in summer to increase the number of locations within 1 km of shore. The weather in winter was highly varied from sunny to heavy rain and light snow. Temperatures ranged -1°C to -7°C.

## 2.3 Oshawa Sites

The Oshawa study area is within Lakeview Park (see Fig. 3), with a few shoreline features such as a beach, a headland to the south-west, a stream with surrounding marshes running into the harbour, and low-rise residential housing. The area contains amble vegetation within parkland and marshland and mature trees line residential roads and backyards. There is some vehicular traffic and foot traffic, especially during spring-fall. Fifteen locations ranging from 11000 m to -143 m from shore were sampled in summer. Lakeview Park Lighthouse sits on a structure that extends into the lake, and so was regarded as negative distance. The weather was consistent during the summer sampling days, sunny to cloudy with mean daily temperatures 16.6°C to 23.2°C. A subset of 10 locations were sampled in the winter period, ranging from 873 m to -34 m from shore. Two new locations (ID 8 and 12) were added in winter to increase the number of locations within 1 km and one location, the Lakeview Park Lighthouse (ID 14), was inaccessible in winter. Lakeview Park Rocks (ID 18) was only sampled on 2 occasions as weather conditions made it unsafe to access. Weather conditions ranged from sunny to light snow with temperatures -1°C to 6°C. Sampling was done on 7 days in both summer and winter.

## 2.4 Sites Parallel to Shore

Six locations from Scarborough, in eastern Toronto, to Oshawa were chosen to measure ozone concentration gradients parallel to the coast. The locations are identified in Fig. 4. Five of the six locations were at commuter train stations: Oshawa, Whitby,

Ajax, Pickering, and Rouge Hill. The stations were isolated from nearby residential housing that were mostly low to medium density. There was regular train and vehicular traffic. The sixth site was at the University of Toronto Scarborough Campus (ID 24). The campus is located north of Highland Creek, and contains river and parkland features, along with low to medium rise buildings and some small pockets of woodlots and shrubbery. There was regular foot and vehicular traffic mainly at the intersection and parking lots. Distance from shore ranged from 3640 m (ID 24) to 70 m at Rouge Hill station (ID 23). Six days were sampled in summer with weather mostly sunny to cloudy and temperatures between 24°C and 30°C. Full information about location sites is provided in Supplemental Information Table S1.

**2.5 Instruments and Measurements**

Ground-level ozone was measured using an Aeroqual 500 handheld monitor. The Aeroqual 500 is a commercial ozone monitor with a gas sensitive semiconductor sensor, with a stated detection limit of 0.001 ppm, and a measurement error ± 0.001 ppm, according to manufacturer's calibration. The monitor was held perpendicular to the wind direction 1.5 m high above ground elevation. Five consecutive values at 1 min intervals were recorded and averaged to generate a single reading. All measurements were made in the afternoon between 12:00 and 20:00 EDT except for those made on July 14[th], 2022, when measurements were made up to 21:00 EDT. This was done to measure where ozone levels are highest during the day for instrumentation reasons and to reduce the influence of diurnal ozone trends as changes are also minimized during peaks. All measurements were made within an approximate 4-hour timeframe on a single day. Nearshore sites under 1 km on a single day were measured within an approximate 1.5-hour timeframe. Wind speeds and wind chill temperature were measured using an AOPUTTRIVER 816B handheld anemometer at the same time as the ozone measurements. Wind direction was measured using a digital compass. GPS information of each sampling site, including longitude, latitude, and elevation, were obtained using the GPS essentials phone application. Hourly wind speed, wind direction, and temperatures were obtained from Environment Canada from https://climate.weather.gc.ca/historical_data/ search_historic_data_e.html from the Toronto City Centre (ID 6158359), Toronto Buttonville Airport (ID 6158409) and Oshawa (ID 6155875) stations. Weather conditions at each site were also recorded from personal observations.

The Aeroqual 500 monitor output was compared with the real-time unverified data from Ontario air quality stations in Oshawa and Downtown Toronto. An average of 10 measurements were taken immediately outside the provincial air quality stations and compared with unverified data from the nearest hour. Monitor measurements agreed with station measurements within 1 standard deviation (see Tables S2 and S3).

Hourly $O_3$ and $NO_2$ data were extracted from Ontario's Ambient Air Monitoring Network in the Oshawa and Toronto Downtown stations from http://www.airqualityontario.com/. Hourly data from January 2021 and July 2021 was extracted on June 22, 2022, and used as a baseline representation of both cities for winter and summer respectively (see Tables S4 and S5).

**2.6 Data Treatment**

The average ozone concentration, expressed as ppbv, and its standard deviation were calculated for each location for each measurement day. Summer and winter values for each city on each measurement day were separately plotted against distance from shore, with a linear least-squares regression applied to all data points within 1 km of shore. Data measured in Toronto on December 10th, 2022, was removed as an outlier from subsequent statistical treatment due to low $R^2$. Slopes were averaged for each city and each season and a two-way ANOVA was applied to average ozone concentration as a function of season and city. Ozone values from the transects parallel to shore in summer were plotted against distance from shore and against distance from Oshawa with Oshawa station (ID 19) set as zero point. Linear regression was applied to these data to determine any gradient present. Wind direction and speed were plotted on a wind rose diagram and a Pearson r correlation and linear regression was applied to determine whether any relationship existed between transient wind speed and ozone concentration. Lake-breeze days were assessed in Toronto by the four criteria outlined by Laird et al., (2001) that had previously been used by Wentworth et al., (2015) to identify lake-breeze days in Toronto. Offshore winds were defined as between 90 - 270 degrees inclusive and onshore winds < 90 degrees and > 270 degrees. Toronto Buttonville airport (ID 6158409) was designated as the inland station and Toronto City Centre (ID 6158359), located on the Billy Bishop Airport island, as on the lake station location. Arithmetic means of daily $O_3$ and $NO_2$ levels were calculated using hourly measurements (12:00 to 21:00 EDT) from the provincial stations in January and July 2021. A two-way t-test was applied on daily mean values between each city in January and July.

**3 Results**

**3.1 Summer Ozone Gradients**

Figures 5 and 6 show a negative linear relationship between ozone concentration and distance from shore in summer on all measurement days for both Toronto and Oshawa. The average slope over the nearest 1 km from shore was -15.4 ± 6.7 ppb/km in Toronto and -23.5 ± 8.5 ppb/km in Oshawa, where the uncertainties given represent one standard deviation from the mean. Toronto gradients are similar to those previously measured in another urban area; Zhang et al., (2020) measured ozone gradients of -15 ppb/km and -18 ppb/km in the north and south shores of Long Island, New York, United States at distances of around 2 km. However, these values are much higher than what Geddes et al., (2021) measured in Boston, Massachusetts, United States where horizontal $O_x$ ($NO_2$ + $O_3$) gradients of 35 and 40 ppb in 15 km, equivalent of 2.3 and 2.7 ppb/km, were observed. The values reported by Stanier et al. (2021) also indicated an equivalent of around -5-6 ppb/km in the west coast of Lake Michigan with 4.1 km. Ozone gradients were greater on average in Oshawa than in Toronto, which also typically showed more elevated levels of ozone nearshore. The Oshawa gradient was smaller than the -37.6 ppb km$^{-1}$ measured by Blanchard & Aherne (2019) in Sandbanks Provincial Park. There was no statistical difference in slope for either Toronto or Oshawa on weekends compared to weekday (two sample t-test, $p > 0.05$).

Similar to results by Blanchard & Aherne (2019), ozone concentrations reached a minimum just under 1 km from shore and then generally plateaued or increased somewhat at distances further from shore. In Toronto, as illustrated in Fig. 5, ozone levels tended to increase moving further inland. This is particularly noticeable on August 13th, 2022, when the 2 points farthest from the lakeshore had concentrations higher than the highest nearshore levels. The observations illustrated in Fig. 6 show that in Oshawa, ozone levels remained constant or continued to decline gently at distances further than ~ 1 km away from shore, up to 11.5 km, and they did not regain the higher levels measured nearshore. This observation may reflect the lack of measurements within the 750 m – 2000 m region or be due to a lack of source emissions to generate more ozone in the daytime in areas not influenced by lake breeze.

Despite the differences in nearshore trends, there were no differences in regional ozone levels between the two cities. Pre-campaign, daily ozone averages from Ontario's Ambient Air Monitoring Network in July and January 2021 were compared and no significant difference was found in summer (see Table S4). To test this, ozone was measured at sites parallel to the shore, shown in Fig. 4, and an average increase $0.26 \pm 0.28$ ppb/km (1 standard deviation) was measured, negligible compared to changes perpendicular to the shore. See supplemental information Fig. S1 and S2 for further information.

The Toronto and Oshawa locations were sampled twice each in summer 2023 to confirm if the trends reported in Fig. 5 and Fig. 6 are consistent (see Fig. S3 and S4). All days displayed a negative linear relationship between ozone concentration and distance from shore. Slopes calculated from linear regression lay in the range seen in 2022 and were within two standard deviations of the means from those measurements, showing a consistency year to year.

### 3.2 Winter Ozone Gradients

Figures 7 and 8 show that in winter, ozone gradients within 1 km of the shoreline were again consistently observed in both Toronto and Oshawa. In Toronto, the steepest slope was observed on Dec 04, 2022, -19.5 ppb/km, approximately twice the value of all other days (see Table S6). The steepest slope in Oshawa was also observed at a similar time on December 02, 2022 (see Table S7). The lowest slope was in Toronto was measured on December 10th, 2022, with poor fit ($R^2 = 0.00013$) and removed from further data analysis. Weather and temperature on these days were not noticeably different from the other days that can be attributed.

Ozone gradients were higher in Toronto than in Oshawa, the opposite of what was observed in summer. The average slope was -16.7 ± 7.3 ppb/km in Toronto and -8.1 ± 5.1 ppb/km in Oshawa, where the uncertainties given again represent one standard deviation from the mean. The range of ozone values was similar for both cities, around 25-45 ppb at the lakeshore (distance to shore = 0 m), unlike in summer where higher maximum values were observed in Oshawa right at the shore.

## 4 Discussion

### 4.1 Lake Breeze and Ozone Gradients

Lake-breezes are known to influence shoreline ozone pollution levels (Cleary, Dickens, et al., 2022) and are common during the summer in the Great Lakes region. Lake breezes were observed 74% of summer days 2010-2012 in Lake Ontario (Wentworth et al., 2015) and over 90% of study days in the southern Great Lakes region during the BAQS-Met study in summer 2007 (Sills et al., 2011). A slight positive relationship in Toronto in summer between wind speed and ozone concentration was observed using Pearson's correlation ($r = 0.45$, $p < 0.001$) and linear regression ($R^2 = 0.20$) (Fig. S5). A

stronger positive relationship was observed in Sandbanks Provincial Park in summer ($R^2 = 0.84$) (Blanchard & Aherne, 2019). There was no relationship in winter between wind speed and ozone concentration in either Oshawa (Pearson's correlation coefficient $r = -0.047$, $p > 0.05$) or Toronto (Pearson's correlation coefficient $r = 0.079$, $p > 0.05$).

Lake-breeze days in Toronto summer 2022 were determined using the criteria outlined by Laird et al., (2001) and were positive for all days except for August 2, 2022. Similarly, the wind rose of Toronto summer 2022 displayed in Fig. 9a also shows an

onshore wind typical of lake breezes from a predominantly SW-SE direction (69% observed) from all wind measurements directions observed during sampling. The only negative result for wind-breeze, August 2[nd], 2022, had the lowest measured slope, -8.6 ppb/km, and wind directions recorded along each sampling site were sporadic with no trend. However, a z-score of + 0.928 and Grubbs' outlier test ($a = 0.050$) shows that this is not a significantly lower value. The steepest negative gradient was measured on August 13[th], 2022, with winds present over a wide range of directions: SW-SE, W, and E, at various sample

sites. Wind measurements were not made during ozone sampling in Oshawa summer 2022 and the lack of available historical climate data prevented the assessment of lake-breeze days in Oshawa.

All days in Toronto winter were negative for lake-breeze except for February 12[th], 2023. This is expected as in winter circulation patterns are reversed with daytime land-breezes leading to a convergence, a net inflow of air, over the lake (Passarelli & Braham, 1981). Offshore winds were observed in Toronto, strongly SW (31% occurrence) and E-NE (28%), as

well as in Oshawa where winds were evenly split SW-N (77%) and SE (17%) (Fig. 9b and c). On February 12[th], the slope was on the steeper end but within the range of days with no lake effect. There are currently only a limited number of studies focused the connection between local scale circulation and ozone or other pollutants in winter and this should be further investigated.

### 4.2 Seasonal Changes in Ozone Gradient

A year-round negative linear relationship between ozone and distance from lakeshore was observed in Toronto and Oshawa.

Figure 10 shows that median and mean slopes of the nearshore ozone gradient in Oshawa were smaller in winter compared to summer. The spread of data is smaller in winter, with outliers on the earliest sampling date Dec 04, 2022, below the minimum, and the latest sampling date February 20[th], 2023, above the maximum. The median and mean slopes in Toronto did not vary seasonally. A two-way ANOVA was performed to compare the effects of season and city on mean slope. There was no

statistically significant difference between the mean gradients observed in the two cities for each season ($p = 0.926$) There was a statistically significant change in slope within Oshawa between summer and winter ($p < 0.001$) and no seasonal difference in Toronto ($p = 0.767$). This seasonal difference is further supported by the sampling completed in summer 2023. Oshawa slopes were once again at similar levels to summer 2022 while Toronto remained consistent (see Fig. S3 and S4).

Ozone gradients observed year-round despite a lack of lake-breeze circulation and relationship with wind speed suggests local circulation is the major contributing factors. One possibility is that there is a "baseline" gradient that occurs due to dilution of air from the increased mixing heights that occurs as the boundary layer gains altitude inland from the coast (Cleary, Dickens, et al., 2022; Loughner et al., 2016; Stroud et al., 2020). A study in Japan reported that in winter and early spring, daytime $O_3$ is described more by the development of a mixing layer and the emission strength of NO near the observation site rather than sea-breeze circulation (Mizuno & Yoshikado, 1983). Loughner et al., (2016) found that the planetary boundary layer in Chesapeake Bay increased from 1.5 km to almost 2.4 km within 5 km inland then plateaued, and that coincided with declines in nitrogen dioxide surface concentration. For Toronto, Stroud et al., (2015) modelled an ozone turbulent mixing height of only a couple of meters for air masses over the lake; this mixing height rapidly grew to become up to 2.3 km altitude after 1 km inland. This distance scale for boundary layer growth is certainly consistent with the results observed in this study, where the lake-edge removal effect was observed within 1 km and ozone concentrations increased or plateaued after this distance. This is also seen within literature where similar steep ozone gradients in measurements at 1-2 km distances from shore to what we report: -18 ppb km$^{-1}$ and -15 ppb km$^{-1}$ by Zhang et al. (2020), -37.6 ppb km$^{-1}$ by Blanchard & Aherne (2019). At farther distances, the gradient is less steep or even rises: -2.3 ppb km$^{-1}$ and -2.6 ppb km$^{-1}$ by Geddes et al., (2021), -4 ppb km$^{-1}$ by Blanchard & Ahrene (2019).

Dilution caused by boundary layer growth, however, does not explain the differences between Toronto and Oshawa or their seasonal differences. Similar gradients should occur in Toronto and Oshawa due to their geographic proximity, and a less steep gradient would be expected in winter, but this is only observed in Oshawa and not in Toronto. It's possible that more mixing occurs in Toronto due to increased turbulent mixing caused by urban infrastructure. Seasonal impacts of temperature and sunlight on ozone concentration are also possible, and we observe an overall decline in ozone mixing ratios in winter, but once again would be expected to influence both cities in a similar manner. Additional small-scale modelling would be needed to determine if the seasonal boundary layer changes are not uniform between the cities.

The significant seasonal variation in ozone gradients observed in Oshawa but not in Toronto suggests vegetation as a key controlling mechanism for the lake-edge removal effect. Comparing summer values among Toronto, Oshawa and Sandbanks, the average ozone gradient away from shore increases going from less to more vegetated areas: $-15.4 \pm 6.7$ (1 standard deviation) ppb/km in Toronto, $-23.5 \pm 8.5$ (1 standard deviation) ppb/km in Oshawa, and -37.6 ppb/km in Sandbanks Provincial Park. Vegetation plays an important role in regulating tropospheric $O_3$, providing both a removal mechanism via uptake to

leaves and in canopy chemistry, and a source, via VOC production, in addition to indirect effects on air chemistry by cooling and shading (Fitzky et al., 2019). Previous studies have reported a forest edge effect, where ozone concentration was reduced within forests compared with open regions, through increased rates of stomatal uptake and dry deposition overwhelming ozone replacement from horizontal wind or higher air layers (Karlsson et al., 2006). Blanchard & Aherne (2019) hypothesized that fragmented vegetation at the Sandbanks site increased removal effects as polluted air masses were transported onshore from Lake Ontario. The seasonal factors may enhance one another, as a study has reported the removal of $O_3$ by vegetation can be enhanced by the presence of sea-breeze through increased relative humidity and turbulent mixing at the surface (Li et al., 2019).

The seasonal stability of the ozone gradient in Toronto suggests a different regime not affected by seasonal changes. Lake-breeze was observed on most sampling days in summer and not present on most days in winter, yet there was no significant difference in the ozone gradient between seasons. Using data from Ontario's Air Quality Monitoring system, average daily $NO_x$ levels in 2021 during sampling hours of 12:00-21:00 EST were higher in Toronto than Oshawa and this difference was statistically significant ($p < 0.05$) for both seasons (see Table S3). Net chemical loss by titration of $O_3$ + NO is known to decrease ozone levels in urban areas compared to nearby rural regions (Cleary, Dickens, et al., 2022). $NO_2$ was not measured in this report, however, other studies on coastal regions have reported this is not a contributing factor as $NO_2$ along with other gas phase compounds and aerosol also decline with distance from shore, and most strongly in the first 1-2 km, due to growth in boundary layer height (Klingberg et al., 2012; Loughner et al., 2016). Elevated levels of NOx over water from ships are also not expected as Toronto is not a major port city. Strong SW winds (Figure 9b) in winter could suggest pollutants being moved along the coast from nearby in industrial areas upstream that contribute to higher ozone levels near the coast that are not present in Oshawa. Further study would be required to tease out the underlying reasons for the seasonal city differences.

**5 Conclusion**

Ground-level ozone gradients perpendicular to Lake Ontario were investigated in Toronto and Oshawa. A linear ozone gradient with respect to distance from shore was consistently observed in both cities throughout the year that included a lake-edge removal effect, consisting of an ozone minimum around 600-800 m from the shore. Local landform and subsequent ozone production regime are connected to ozone gradients as they increased in steepness from urban to rural, $-15.4 \pm 6.7$ (1 standard deviation) ppb/km in Toronto, $-23.5 \pm 8.5$ (1 standard deviation) ppb/km in Oshawa, and $-37.6$ ppb/km in Sandbanks Provincial Park (Blanchard & Aherne, 2019). A slight urban-rural ozone gradient was also observed in summer with ozone levels increasing towards more urban regions.

The seasonal changes in the lakeshore ozone gradient in Oshawa and Toronto suggest that the lake-edge removal effect is less strongly associated with lake breeze circulation than previously assumed. A year-round lake-edge removal effect, the steep

gradient within 1 km, is hypothesized to be the result of the growth of boundary layer height diluting the air moving inland from the coast. Average ozone gradients were smaller in winter than summer in Oshawa while they remained similar in Toronto. In addition, the gradient remained present in winter despite the lack of lake-breeze and there was no correlation with local wind speed in winter in either city. Lake-breeze circulation and deposition by vegetation are hypothesized to cause the stronger gradients in Oshawa summer as well as the strong seasonal difference seen there compared to Toronto. In Toronto, increased turbulent mixing due to infrastructure could also be minimizing the effects of lake-breeze circulation in summer. Although differences in how the boundary layer height increases perpendicular to the lakeshore are not expected due to the proximity of the two cities, they may indeed play a role in why the gradient does not change seasonally in Toronto. This possibility awaits regional modelling studies for its testing. Further studies to co-monitor $O_3$ and $NO_x$ gradients perpendicular to the shore could elucidate the mechanism within urban areas.

## Data availability

The data used in this paper can be obtained on request from the corresponding author (james.donaldson@utoronto.ca).

## Author contributions

Both authors designed the field campaign and YH carried out the field measurements. YH completed the data analysis, and both authors discussed the data and findings. YH prepared the draft manuscript. Both authors reviewed and approved the final version of the manuscript.

## Competing interests

The authors declare that they have no conflict of interest.

## Acknowledgements

Thank you to Ontario Tech University for access to campus grounds to the Oshawa air quality monitor. This work was supported by the University of Toronto (through a Centre of Global Change Science summer fellowship to YH) and by NSERCC.

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

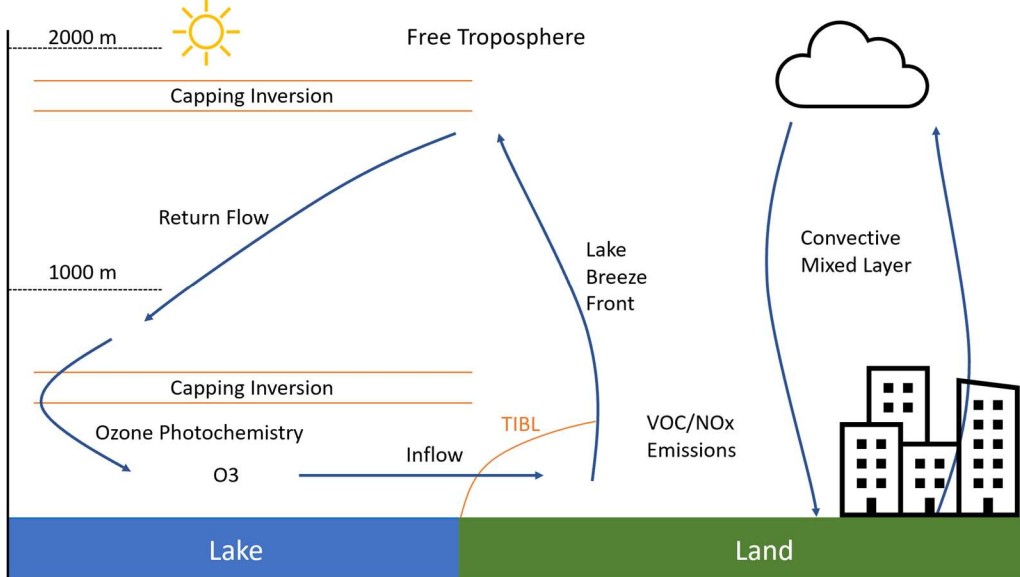

**Figure 1: Simplified model of the connection between lake breeze circulation and ozone. Ozone precursors are trapped within the stable inversion layer over the water and react in the day to produce O3 that is moved inland by lake breeze. Arrows depict motion of air masses. The thermal internal boundary layer (TIBL) is shown in orange and grows in height with distance inland until it reaches the height of the convective mixed layer. Adapted from Wentworth et al. (2015) and Stroud et al. (2020).**


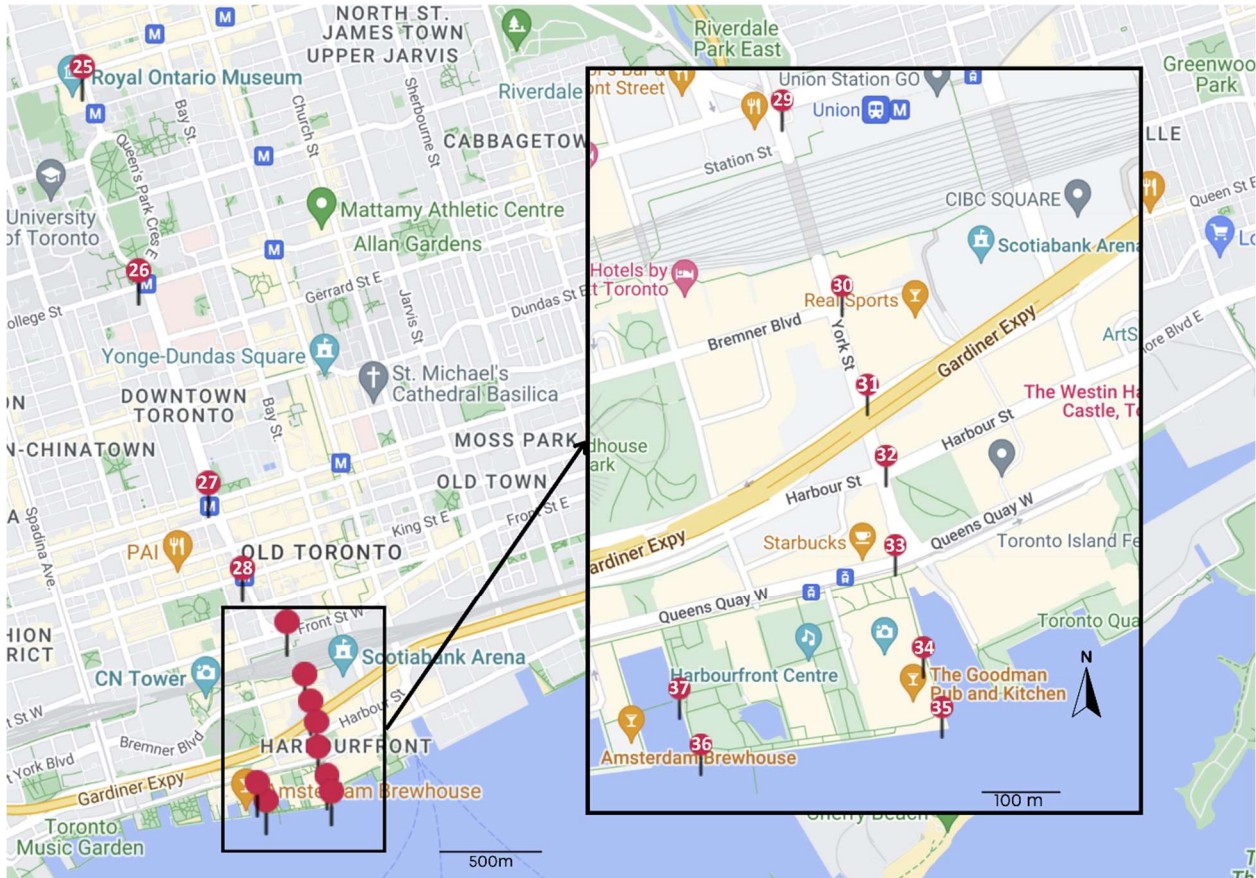

**Figure 2: Ozone sampling locations in Toronto for transects perpendicular to shore. © Google Maps 2023.**


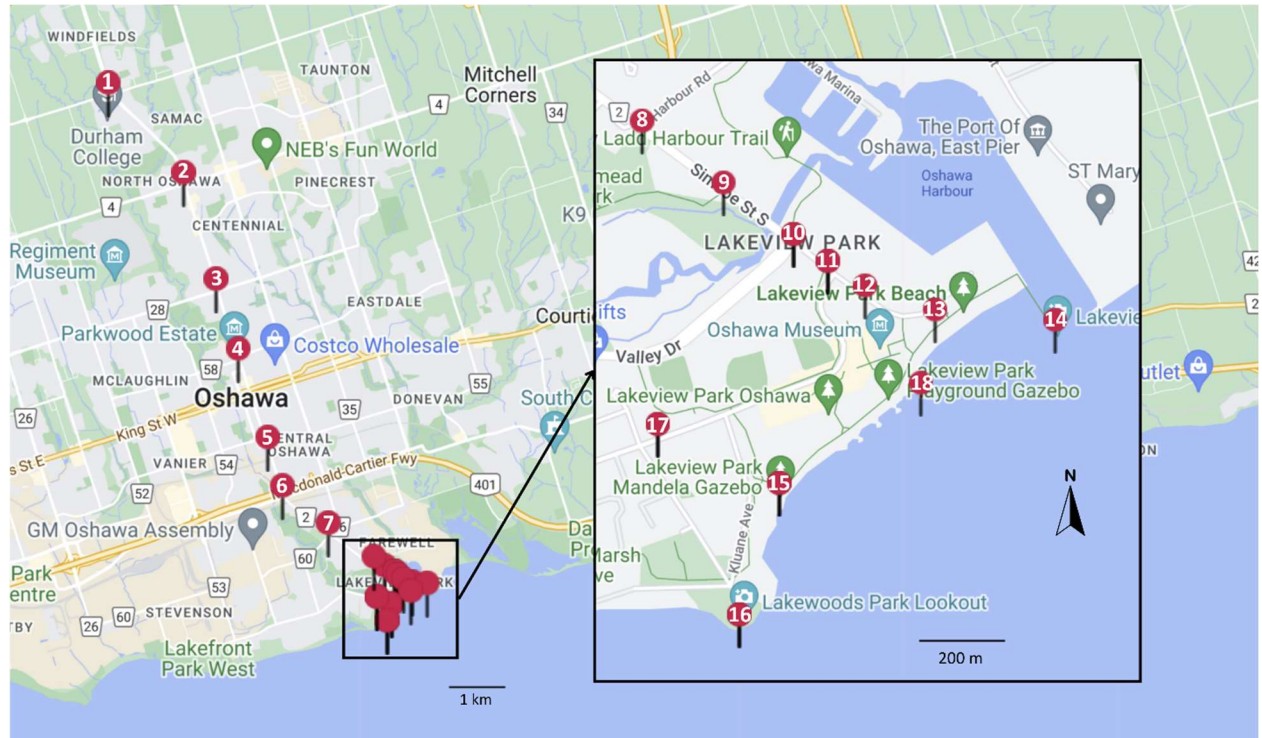

**Figure 3: Ozone sampling locations in Toronto for transects perpendicular to shore. © Google Maps 2023.**

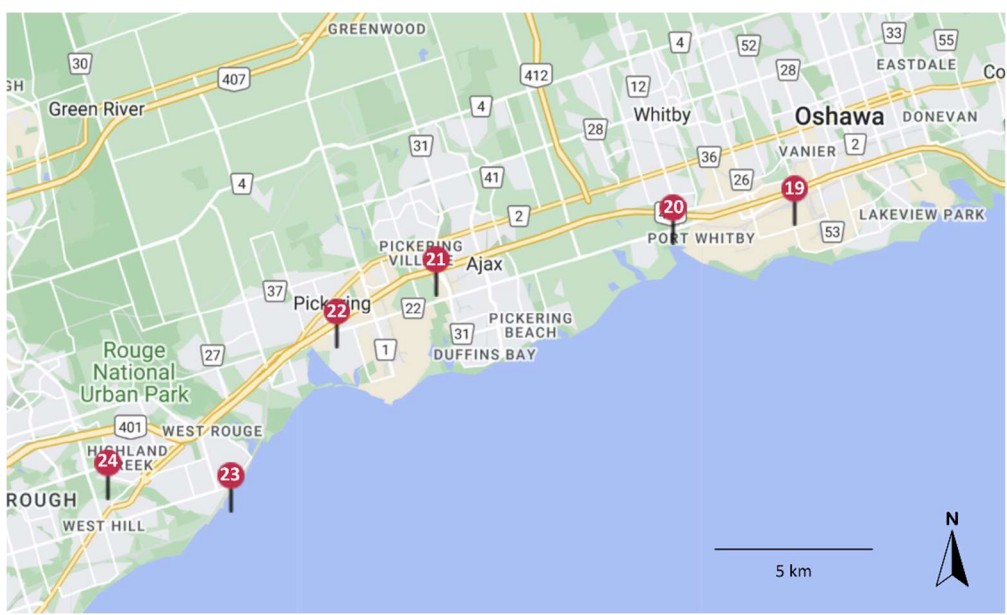

**Figure 4. Ozone sampling sites in summer 2022 for parallel transect. © Google Maps 2023.**


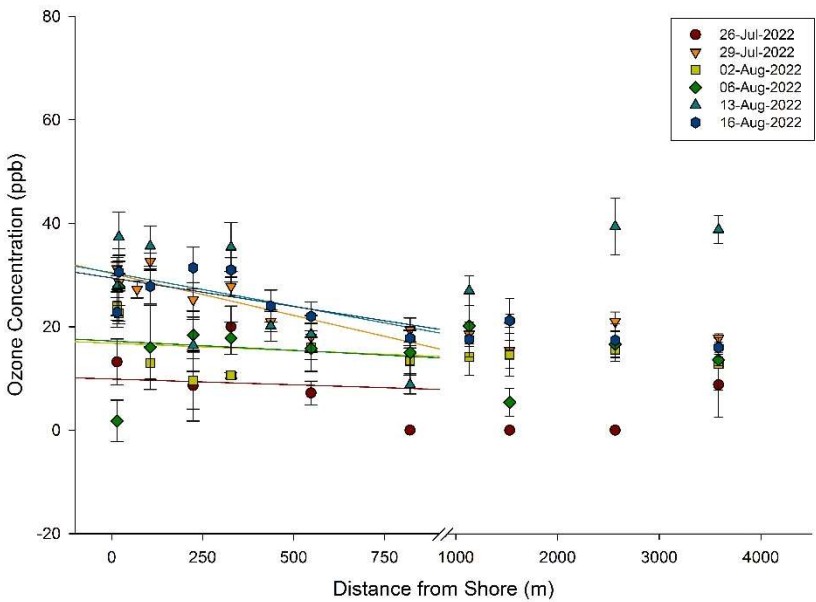

**Figure 5. Scatter plot of average O$_3$ (ppb) to distance from the shore of Lake Ontario (m) in Toronto summer with linear regression within 1 km. Error bars represent ± 1 standard deviation from the mean of five consecutive 1 min measurements.**

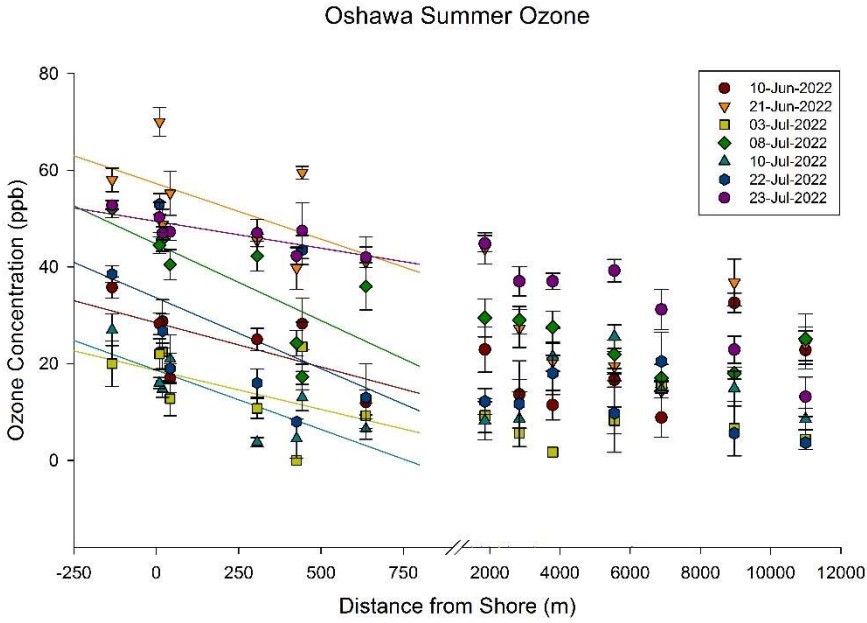

**Figure 6. Scatter plot of average O$_3$ (ppb) to distance from the shore of Lake Ontario (m) in Oshawa summer with linear regression within 1 km. Error bars represent ± 1 standard deviation from the mean of five consecutive 1 min measurements.**

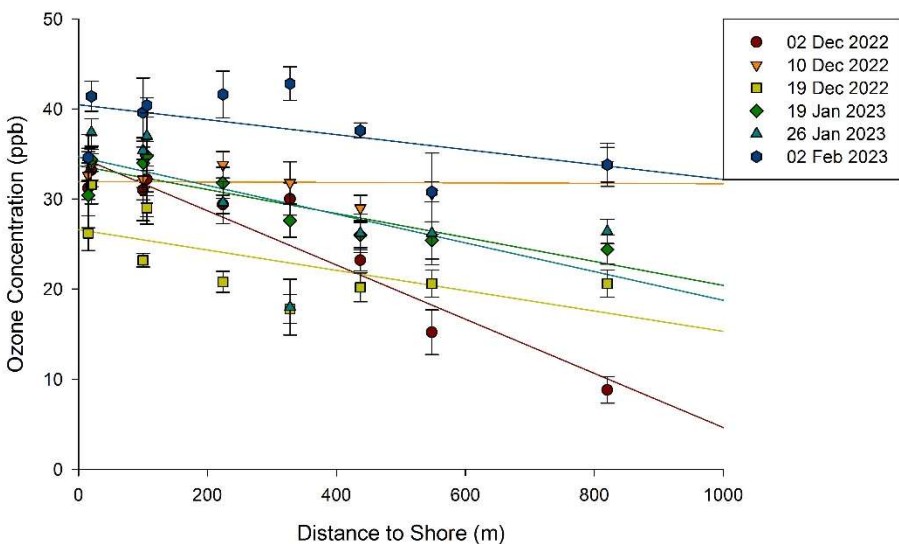

Toronto Winter Ozone

**Figure 7. Scatter plot of average O₃ (ppb) to distance from the shore of Lake Ontario (m) in Toronto winter with linear regression within 1 km. Error bars represent ± 1 standard deviation from the mean of five consecutive 1 min measurements.**

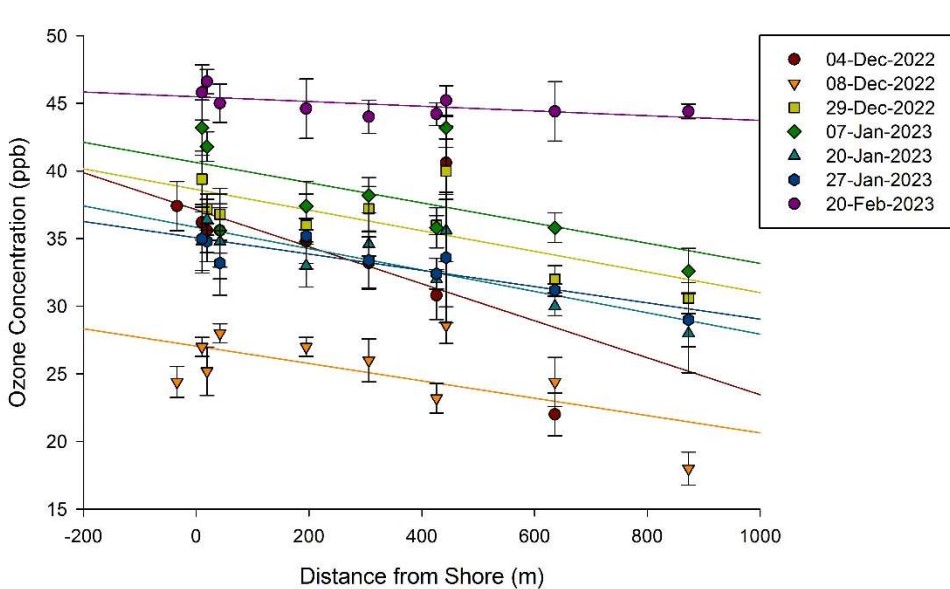

Oshawa Winter Ozone


**Figure 8. Scatter plot of average O₃ (ppb) to distance from the shore of Lake Ontario (m) in Oshawa winter with linear regression within 1 km. Error bars represent ± 1 standard deviation from the mean of five consecutive 1 min measurements.**

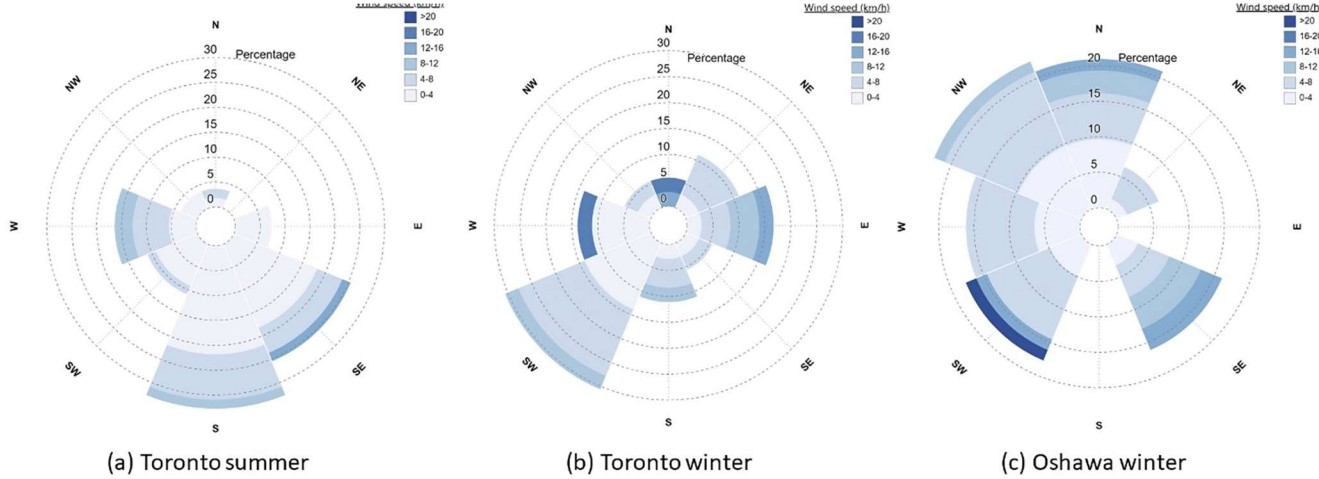

**Figure 9. Wind rose plots displaying the relative frequencies of wind direction (cardinal direction) and speed (km/h) at sampling sites in (a) Toronto summer, (b) Toronto winter, and (c) Oshawa Winter. Data was not collected during Oshawa summer. Made with www.WindRose.xyz.**

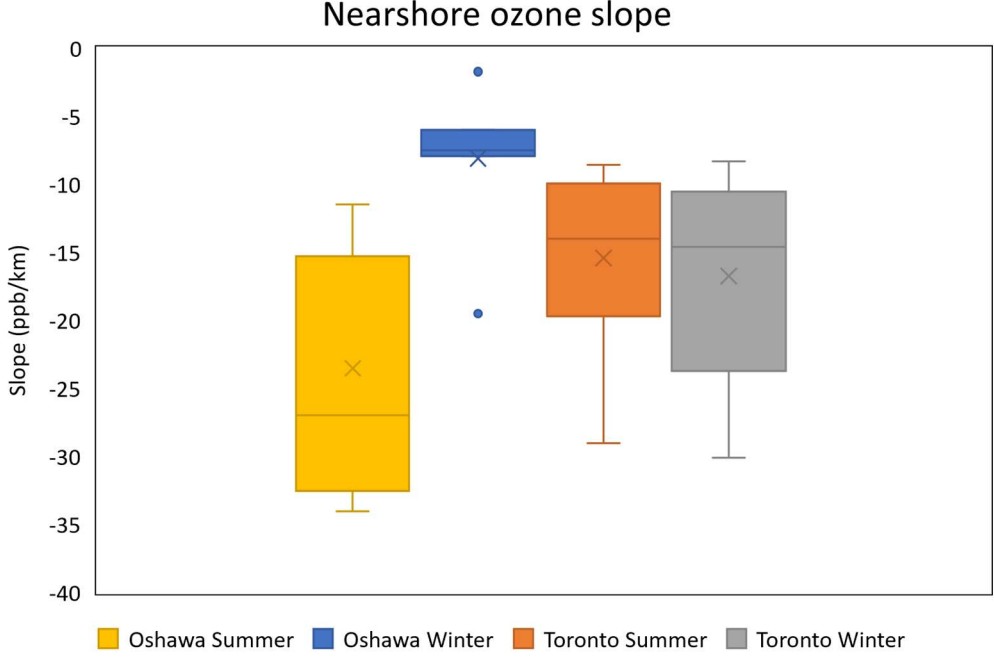

**Figure 10. Box-and-whisker plot of median nearshore slopes in Oshawa and Toronto in two seasons, summer (Jun-Aug) and winter**
**(Dec-Feb). The edges of the box represent the interquartile range, the median of the upper and lower half of the data exclusive of the median. The error bars represent the maximum and minimum values. The horizontal line within the box shows the value of the median and the x symbol shows the mean. Individual dots represent outliers.**