# Peer review of "Table S1. Site ID, name, location (latitude and longitude), elevation (m), distance from Oshawa (m), distance from shore (m) for the 37 locations selected for ozone sampling in Oshawa and Toronto, Ontario. Distance from Oshawa is measured from ID 19. Sites measured in each season are marked with a"

_EGUsphere, 2023_

## Author Comment (AC1)

Response to Reviewer #1

We want to thank Reviewer 1 for their careful reading of the manuscript and the useful and insightful suggestions. The revised version has been substantially improved, we think, by addressing these concerns. The major suggestion was that we should properly consider local meteorological effects in our discussion of the steep near-shore ozone gradients. We have done so in the revised version, with a substantially rewritten discussion and some further clarification of our measurements. We now conclude that a near-shore concentration gradient is to be expected (and is generally seen) due to the rapid increase in the boundary layer height, moving inshore from the lake (or an ocean). This is further expected due to the "lake breeze" effect, generally observed during the summer measurement period. Interestingly, there is no such lake breeze in winter, where the gradients persist. The differences we observe between the Oshawa and Toronto gradients, and the seasonality we see in Oshawa, are perhaps best explained by differences in the ozone dry deposition rates near the shorelines.

We have included sea breeze along with lake breeze effects in our analysis, augmenting the citation list accordingly.

Below we list the specifics of the changes to the manuscript in the revised version.

Major comments:

- Many lake breeze related papers were cited, but papers relating to sea and bay breezes, which are essentially the same phenomenon as lake breezes, were ignored. I suggest the authors provide more background material relating to recent studies involving gradients in air pollution near coastal areas.
    - Added to introduction references to studies by Han et al., (2023) in Hangzhou, China, Zhang et al., (2020) in Long Island, New York, United States, Finardi et al., (2018) in Naples, Italy, and Stauffer & Thompson, (2015) in Chesapeake Bay, United States.
    - Added in results section comparisons to measured ozone gradients at land-sea boundaries.
        - Zhang et al., (2020) in Long Island, New York, USA measured surface ozone gradients of 18 ppb $km^{-1}$ and 15 ppb $km^{-1}$ respectively on two case study days in around 2 km distance.
        - Geddes at al., (2021) measured gradients of Ox (NO2 + O3) greater than 30 ppb in 15 km during sea breeze days in Boston, New England, United States.

Han, Z. S., Liu, H. N., Yu, B., & Wang, X. Y. (2023). The effects of coastal local circulations and their interactions on ozone pollution in the Hangzhou metropolitan area. Urban Climate, 48, 101417. https://doi.org/10.1016/j.uclim.2023.101417

Geddes, J. A., Wang, B., & Li, D. (2021). Ozone and Nitrogen Dioxide Pollution in a Coastal Urban Environment: The Role of Sea Breezes, and Implications of Their Representation for Remote Sensing of Local Air Quality. Journal of Geophysical Research: Atmospheres, 126(18). Scopus. https://doi.org/10.1029/2021JD035314

Zhang, J., Ninneman, M., Joseph, E., Schwab, M. J., Shrestha, B., & Schwab, J. J. (2020). Mobile Laboratory Measurements of High Surface Ozone Levels and Spatial Heterogeneity During LISTOS 2018: Evidence for Sea Breeze Influence. Journal of Geophysical Research: Atmospheres, 125(11), e2019JD031961. https://doi.org/10.1029/2019JD031961

Finardi, S., Agrillo, G., Baraldi, R., Calori, G., Carlucci, P., Ciccioli, P., D'Allura, A., Gasbarra, D., Gioli, B., Magliulo, V., Radice, P., Toscano, P., & Zaldei, A. (2018). Atmospheric Dynamics and Ozone Cycle during Sea Breeze in a Mediterranean Complex Urbanized Coastal Site. Journal of Applied Meteorology and Climatology, 57(5), 1083–1099. https://doi.org/10.1175/JAMC-D-17-0117.1

Stauffer, R. M., & Thompson, A. M. (2015). Bay breeze climatology at two sites along the Chesapeake bay from 1986–2010: Implications for surface ozone. Journal of Atmospheric Chemistry, 72(3), 355–372. https://doi.org/10.1007/s10874-013-9260-y

- Is the "lake-edge removal effect" a common term? If not, I recommend not using it. It sounds like the lake edge is removing ozone, but I don't think that is what is happening.

  - This term was used, and seems to only have been used, by Blanchard & Aherne (2019) to describe the steeper gradient that occurred around 1 km in additional to a general gradient that occurs with distance to shore at greater resolutions > 1 km.

  - We used this term to differentiate between the two observations and believe it sufficiently describes the removal of ozone observed.

Blanchard, D., & Aherne, J. (2019). Spatiotemporal variation in summer ground-level ozone in the Sandbanks Provincial Park, Ontario. Atmospheric Pollution Research, 10(3), 931–940. https://doi.org/10.1016/j.apr.2019.01.001

- The authors hypothesize that the observed ozone gradient is primarily due to deposition and chemistry. There are other contributing factors. I strongly suggest the authors review the following paper that discusses sharp gradients in concentrations and deposition of nitrogen species along coastlines (Loughner, C.P., M. Tzortziou, S. Shroder, and K.E. Pickering (2016), Enhanced dry deposition of nitrogen pollution near coastlines: A case study covering the Chesapeake Bay estuary and Atlantic Ocean coastline, Journal of Geophysical Research – Atmospheres, 121, 14,221-14,238.). The gradient in boundary layer height near the coastline may be a large contributing factor in the observed ozone gradient, but boundary layer height was not mentioned in the manuscript. While ozone titration might be occurring, there is no evidence provided that it is.

  - Ozone titration hypothesis has been removed due to lack of measurement of $NO_2$ and other means of evidence. In particular, we did not measure ozone in winter further than 1 km distances to confirm that levels in Toronto will increase again.

  - We discuss the impact of boundary layer height changes near the coastline in relation to ozone gradients and include the paper listed. We will also include the boundary layer height changes in Toronto as modelled by Stroud et al., (2020)

    - On 28 July 2015 around 5:00 PM local time, the mixing length at the surface increases from around 1-3 m on the lake to 70-100 m in around 1 km distance in downtown Toronto. After this point, the model shows a uniformly well-mixed convection up to 2.3 km altitude. This suggests the origin of the steeper

gradient or lake- edge removal is largely influenced by this growth in boundary layer height.

- o However, assuming the boundary layer height changes uniformly along the lake, this does not account for the differences between cities. Seasonal changes in boundary layer height should also reduce the ozone gradient in winter for both cities and is not observed in Toronto. We hypothesize that changes in deposition are related to the differences in ozone gradient between the two cities and their seasonal changes in ozone gradient slope.

Stroud, C., Ren, S., Zhang, J., Moran, M., Akingunola, A., Makar, P., Munoz-Alpizar, R., Leroyer, S., Bélair, S., Sills, D., & Brook, J. (2020). Chemical Analysis of Surface-Level Ozone Exceedances during the 2015 Pan American Games. Atmosphere, 11(6), 572. https://doi.org/10.3390/atmos11060572

- • I think the analysis would benefit if the analyzed gradients in observed ozone concentrations were performed separately based on wind direction (onshore vs offshore flow). When the winds are onshore, the gradient may primarily be due to the gradient in boundary layer height. When there is offshore flow, the gradient may be just due to the coastline being downwind of emissions of ozone precursors from the urban area.

  - o Wind direction was presented using wind rose plots for summer and winter in Toronto and Oshawa where data was collected. The presence of onshore flow in summer versus not present or offshore in winter was discussed.
  - o We include assessment of lake-breeze days following criteria by Laird et al., (2001) that has also been previously used by Wentworth et al., (2015) to identify lake-breeze circulation in Toronto.
    - ▪ All days in Toronto were positive for lake-breeze except for August 2, 2023. This is incidentally also the day when the slope was the lowest, -0.0086 ppb/m, and our own measured wind directions were sporadic with no trend.
    - ▪ A z-score of + 0.928 and Grubbs' outlier test (a = 0.050), however, does not show that this is a significantly lower value.
    - ▪ The same program was also run on Toronto winter values that resulted in one positive for February 12, 2023. The slope was on the steeper end but not the highest recorded.
  - o We also assessed lake-breeze in Oshawa using the same method. No appropriate lake meteorological station with accessible data was available so the same station in Toronto was used.
    - ▪ All days showed lake-breeze except for June 10th, 2022. The slope on this day was below the mean but not the minimum.
    - ▪ Winter had no positives.

Laird, N. F., Kristovich, D. A. R., Liang, X.-Z., Arritt, R. W., & Labas, K. (2001). Lake Michigan Lake Breezes: Climatology, Local Forcing, and Synoptic Environment. Journal of Applied Meteorology, 40(3), 409–424. https://doi.org/10.1175/1520-0450(2001)040<0409:LMLBCL>2.0.CO;2

Wentworth, G. R., Murphy, J. G., & Sills, D. M. L. (2015). Impact of lake breezes on ozone and nitrogen oxides in the Greater Toronto Area. Atmospheric Environment, 109, 52–60. https://doi.org/10.1016/j.atmosenv.2015.03.002

- Combining this analysis with PBL height calculated from a NWP model would benefit this manuscript.
  - As mentioned previously, boundary layer from Toronto is inferred from model results from the Stroud et al., (2020) paper.

Minor comments:

- Lines 9-10: In addition to considering removing "lake-edge removal effect" here and throughout the paper (see comment above), I suggest changing "where ozone concentration decreases within the first 500 m to 1 km perpendicular to the lake" to "where ozone concentration decreases with distance from the lake within the first 500 m to 1 km" to make sure the reader understands you see this gradient near the coastline onshore and not just offshore.
- Line 27: change "airflow moving" to "airflow near the surface moving"
- Lines 30-31: end sentence after "lake" and delete the remainder of the sentence.
- Line 31: This sentence refers to a land breeze the figure shows a lake breeze.
- Line 32: change "ozone concentration" to "ozone and ozone precursor concentrations"
- Line 33: change "O3 inland" to "O3 and O3 precursors inland"
- Line 47: change "lake-breeze" to "lake-breezes"
- Line 53: change "lake breeze was" to "lake breezes were"
- Line 61: change "further" to "farther"

The revised text has incorporated all these suggestions.

---

## Author Comment (AC2)

**Response to Reviewer #2**

We thank Reviewer 2 for their careful reading of the manuscript. Most of the comments relate to clarifying aspects of the original version; we have addressed all of these in the revised version. The substantive comment regarding seasonal differences in meteorological factors which may impact the measured ozone gradients. Following the suggestions of Reviewer #1, we have made significant changes to the discussion; it now explicitly includes some analysis of boundary layer height changes perpendicular to the shoreline, as well as lake breeze effects.

Our specific responses to the comments of Reviewer 2 are given below.

Line 53: Put a "The" before "Average" or make concentrations plural

- Edited

Line 131: Why were measurements only made during the afternoon? Please clarify.

- Measurements were made in mid day to afternoon, when ozone levels peak, in an attempt to reduce the influence of diurnal ozone trends on observations. Lake-and sea-breeze circulations tend to establish around noon and after as well, and ozone gradients would be strongest. This explanation has been added to the revised text.

Line 145-146: Change "was" to "were", data is considered plural. Check manuscript for other instances.

- Edited

Line 159: For the hourly O3 and NO2 from the provincial stations, are these data only encompassing the hours of 12:00-21:00 EDT, similarly to the Aeroqual 500? It should be noted specifically how the monitoring station data is averaged

- Monitoring data was an arithmetic mean from daily hourly values from 01:00 to 24:00 EDT of a single day. We changed the average to encompassing sampling hours 12:00-21:00 EDT.

Line 265-266: The authors do not mention the shallower mixed layers, decreased photolysis rates of NO2 and overall decreased ozone production rates during winter time. These are possible contributions to the lack of a relationship between wind speed and ozone concentration. Ozone gradients may be more impacted by meteorology during summer, while during winter the regional background and local sources may contribute more. The authors should explore and describe how during winter there is less dynamic variability in ozone due to the possible reasons mentioned above.

- We are adding the boundary layer height changes in Toronto as modelled by Stroud et al., (2020).
  - At around 5:00 PM local time, the mixing length at the surface increases from around 1-3 m on the lake to 70-100 m in around 1 km distance in downtown Toronto. After this point, the model shows a uniformly well-mixed convection up to 2.3 km altitude. This suggests the origin of the steeper gradient or lake-edge removal is largely influenced by this growth in boundary layer height.

- o However, assuming the boundary layer height changes uniformly along the lake, this does not account for the differences between cities.
- An overall shallower mixed layer would result in decreased gradients in both Toronto and Oshawa, but Toronto displayed no difference between the seasons. Decreased ozone production and background levels are somewhat observed in our observations, however, would once again decrease ozone gradients overall but Toronto is not affected.
- We are adding assessment of lake breeze in Toronto summer following criteria by (Laird et al., 2001) that has also been used by (Wentworth et al., 2015) to identify lake-breeze circulation in Toronto.
  - o All days in Toronto from this test showed lake-breeze except for August 2, 2023. This is incidentally also the day when the slope was the lowest, -0.0086 ppb/m, and our own measured wind directions were sporadic with no trend.
  - o A z-score of + 0.928 and Grubbs' outlier test (a = 0.050), however, does not show that this is a significantly lower value.
  - o The same program was also run on Toronto winter values that resulted in one positive for February 12, 2023. The slope was on the steeper end but not the highest recorded.
  - o In Oshawa, all summer days showed lake-breeze except for June 10th, 2022. The slope on this day was below the mean but not the minimum. Winter had no positives.
- From this we hypothesize that the base nearshore ozone gradient is the result of the growth of the boundary layer height moving from lake to land and extends around 1 km inland. Lake-breeze meteorology and seasonal production rates impact the absolute values of ozone measured. Seasonal differences in ozone gradient appear to be impacted more by regional geographic factor.

Stroud, C., Ren, S., Zhang, J., Moran, M., Akingunola, A., Makar, P., Munoz-Alpizar, R., Leroyer, S., Bélair, S., Sills, D., & Brook, J. (2020). Chemical Analysis of Surface-Level Ozone Exceedances during the 2015 Pan American Games. Atmosphere, 11(6), 572. https://doi.org/10.3390/atmos11060572

---

## Author Comment (AC3)

We want to thank both Reviewers for their careful reading of the manuscript and the useful and insightful suggestions. The revised version has been substantially improved, we think, by addressing these concerns. We have posted detailed responses to both as replies to their comments.

The major improvements have been to consider local meteorological effects in our discussion of the steep near-shore ozone gradients. We have substantially rewritten the discussion to do so, and have also added further clarification of our measurements. We now conclude that a near-shore concentration gradient is to be expected (and is generally seen) due to the rapid increase in the boundary layer height, moving inshore from the lake (or an ocean). This is further expected due to the "lake breeze" effect, generally observed during the summer measurement period. Interestingly, there is no such lake breeze in winter, where the gradients persist. The differences we observe between the Oshawa and Toronto gradients, and the seasonality we see in Oshawa, are perhaps best explained by differences in the ozone dry deposition rates near the shorelines.

All minor corrections and suggestions made by both reviewers have been incorporated into the revised version.